# A Performance-Enhanced Liquid Metal-Based Microheater with Parallel Ventilating Side-Channels

**DOI:** 10.3390/mi11020133

**Published:** 2020-01-24

**Authors:** Lunjia Zhang, Pan Zhang, Ronghang Wang, Renchang Zhang, Zhenming Li, Wei Liu, Qifu Wang, Meng Gao, Lin Gui

**Affiliations:** 1Key Laboratory of Cryogenics, Technical Institute of Physics and Chemistry, Chinese Academy of Sciences, 29 Zhongguancun East Road, Haidian District, Beijing 100190, China; zhanglunjia14@mails.ucas.ac.cn (L.Z.); zhangpan18@mails.ucas.ac.cn (P.Z.); wangronghang14@mails.ucas.ac.cn (R.W.); zhangrenchang15@mails.ucas.ac.cn (R.Z.); wangqifu16@mails.ucas.ac.cn (Q.W.); mgao@mail.ipc.ac.cn (M.G.); 2School of Future Technology, University of Chinese Academy of Sciences, Beijing 100039, China; 3Information Science Academy, China Electronics Technology Group Corporation, Beijing 100086, China; 4Energy Storage and Novel Technology of Electrical Engineering Department, China Electric Power Research Institute, Beijing 100192, China; lizhenming@epri.sgcc.com.cn (Z.L.); liuwei3@epri.sgcc.com.cn (W.L.)

**Keywords:** liquid metal-based microheaters, ventilating side-channels, voids, trapped air

## Abstract

Gallium-based liquid metal can be used as a material for microheaters because it can be easily filled into microchannels and electrified to generate Joule heat, but the liquid metal-based microheater will suffer breakage induced by voids forming within the liquid metal when the temperature normally gets higher than 100 °C. To resolve this problem, a novel liquid metal-based microheater with parallel ventilating side-channels is presented. It consists of a liquid-metal heating channel and two parallel ventilating side-channels. The heating channel is connected with the side-channels by small gaps between polydimethylsiloxane (PDMS) posts. Experimental results show that this novel microheater can be heated up to 200 °C without damage. To explain its excellent performance, an experiment is performed to discover the development of the voids within the liquid-metal heating channel, and two reasons are put forward in this work on the basis of the experiment. Afterward pressing and bending tests are conducted to explore the mechanical stability of the novel microheaters. Finally, the microheaters are applied to warm water to show their good flexibility on non-flat surfaces. In consequence, the novel liquid metal-based microheater is believed to be widely applicable to soft micro-electro-mechanical system(MEMS) heating devices.

## 1. Introduction

Recent years have seen great progress in employing microfluidic technology in the field of wearable electronics [1], physiological monitoring systems [2] and biochemical synthesis application [3]. Combining microfluidic technology with liquid metals provides an excellent way to make flexible devices and thus it has become a hot research frontier [4]. Among all liquid metal materials, a eutectic alloy of gallium, indium and tin, EGaInSn (Ga_67_In_20.5_Sn_12.5_, weight percentage, 67% Ga, 20.5% In, 12.5% Sn, melting point 10.5 °C), and a eutectic gallium-indium alloy, EGaIn (Ga_75.5_In_24.5_, weight percentage, 75.5% Ga, 24.5% In, melting point 15.5 °C), have gained increasing attention in recent years due to their low melting point (below room temperature), high surface tension, high electrical and thermal conductivity, and low toxicity [5]. Ga_67_In_20.5_Sn_12.5_ and Ga_75.5_In_24.5_ have shown great potential use in fabricating various soft electronic components [6], such as functional circuits [7], flexible sensors [8], stretchable coils [9], and microheaters [10].

Our study focuses on a liquid metal-based soft microheater, whose heating element is fabricated by injecting the electrically conductive liquid metal into elastomeric microchannels. These microchannels can be patterned onto elastic substrates through micromachining technology such as soft lithography [11], direct writing ablation [12], 3D printing [13], etc. Among many available elastic materials, polydimethylsiloxane (PDMS) is one of the most popular candidates due to its good flexibility and biocompatibility [14]. This work therefore fabricates and optimizes a flexible microheater which adopts PDMS as its soft substrate and utilizes EGaIn-filled microchannels to produce Joule heat. In contrast with a conventional solid-metal microheater, the soft microheater in our work is fabricated much more easily and economically because the conventional microheater often requires complicated vaporing, sputtering or deposition processes to form inertial metal films such as platinum (Pt) or gold (Au) film on the substrates [15]. In addition, the rigid-metal film is vulnerable to mechanical deformation because it may easily crack or be detached from soft substrates upon mechanical stress [16], whereas the liquid metal is more robust to fatigue because it can maintain the liquid state at room temperature. Furthermore, the liquid-metal heating channels can be readily self-aligned with other fluidic channels because all components of the microheater can be co-fabricated, thereby reducing multiple complicated alignment procedures and simplifying the whole fabrication process [17].

Despite excellent mechanical softness and easy fabrication of the liquid metal-based microheaters, they are easy to break when the liquid-metal heating channels are subjected to high temperature. This breakage is caused by voids forming and expanding within the microchannels. So far, two hypotheses have been proposed to account for the formation of the voids: (1) material temperature expansion coefficient (TEC) mismatch effects [18]; (2) metallic electromigration theory [19,20].

Je and Lee explained that the difference in thermal expansion between PDMS and EGaIn resulted in void formation and development at increasing temperature, but their hypothesis is based on the theoretical possibility rather than an experimental investigation [18]; Ma et al. adopted electromigration theory to explain the break-up phenomena of room-temperature liquid metals [19], but because the cracked residue in their experiment is mainly oxide, that might only indicate the voids formed within the solid oxides of GaIn instead of the liquid metal itself. What’s more, Ma et al.’s [19] work didn’t put forward an effective method to solve the problem of voids, so neither hypotheses provides strong evidence to support the explanation for the formation of the voids in liquid metals and none of them gave a perfect solution for it either. The liquid-metal break-up induced by voids severely limits the applications of microheaters, because the voids usually occur at about 50 °C and finally result in breakage when the temperature reaches around 100 °C. Je and Lee [18] partly solved the problem by connecting the GaIn-based microheater with a pneumatic pump to continuously inject the liquid metal into the microchannels. The method effectively filled the voids instantly and generated a uniform temperature field, but it required an extra pump to supply the liquid metal and the liquid metal was not well-sealed, which makes the microheater not handy for use.

In our work, a novel solution to solve this problem of liquid metal microheaters is proposed. A new structure is adopted as the solution. It consists of three parallel (from a top view) microchannels including a heating channel and two ventilating side-channels. We first compare the microheaters with and without the ventilating side-channels in the same environment, the results show that the maximum temperature of the microheater with the ventilating side-channels is up to 223.4 °C, whereas that of the microheater without the ventilating side-channels is only 113.7 °C. Then an experiment was designed to explore the reasons and explained why the ventilating side-channels can notably limit the voids forming and increase the maximum temperature. Afterward, mechanical tests are conducted to quantify the stability of the microheaters under pressing and bending stresses. Finally, the new microheaters were used to heat water from 27 °C to 55 °C in 30 min by bending and sticking them to the bottom and the sidewall surface of a cup. The new microheater is expected to be extensively used in wearable and medical thermal microdevices.

## 2. Experiments and Methods

First, the microstructure of the soft microheater was fabricated on a silicon wafer using SU-8 2075 photoresist (Microchem, Westborough, MA, USA) through soft lithography. After that, liquid PDMS was cast on the wafer and placed on a hot plate at 65 °C for 2.5 h. The microheater mold thus formed following curing of the PDMS, and the PDMS mold was finally peeled off from the silicon wafer.The microstructure of the three parallel microchannels is shown in Figure 1a. It comprises two parallel ventilating side-channels and one main heating channel in the middle. Two rows of micro PDMS posts are fabricated to separate the heating channel from the ventilating side-channels respectively, and the heating channel is still connected with the ventilating side-channels by small gaps between the PDMS posts. The average thickness of the microstructure is about 1500 μm. The PDMS posts are all 100 μm wide and 180 μm long. We then punched holes at both ends of the three microchannels and bonded the microheater mold onto a 600 μm thick PDMS substrate. Finally, EGaIn was injected into the heating channel manually to form soft liquid metal filled microchannel. As shown in Figure 1b, due to high surface tension of EGaIn, the micro PDMS posts can guide the liquid metal to flow through the heating channel without overflowing into the two parallel side-channels by sophisticatedly controlling the gaps between the PDMS posts [21]. The ventilating side-channels are open to the air outside through the punched holes. Finally, silver plated copper wires were inserted into the holes (inlet and outlet) of the heating channel to keep in contact with the EGaIn. S-shaped microchannels with (Figure 1c) and without (Figure 1b) the ventilating side-channels were fabricated for the liquid metal-based microheater. The average thickness of the soft microheaters tested in the experiments is about 2100 μm, if not otherwise mentioned.

## 3. Experimental Principle and Setup

Experiments were performed to compare the highest temperature that the microheaters could reach with/without the ventilating side-channels under the same voltage. By applying voltages to the microheaters, the Joule heat generated can be as expressed in Equation (1):(1)$$P=\frac{U^{2}}{R}$$
where *P* is the power of the microheater, *U* is the voltage applied to the microheater, *R* is the resistance. The microheater was connected with a DC power (DH1720A, DaHua Power, Beijing, China) via the copper wires. A standard K-type thermocouple was placed beneath the central area of the microheater to measure the temperature. And a data acquisition unit (34970A, Agilent, Santa Clara, CA, USA) was used to collect real-time temperature values. The experimental setup for the Joule-heat generation test is illustrated in Figure 2a.

## 4. Results and Discussion

### 4.1. Experimental Results

The performance of the microheaters with and without the parallel ventilating side-channels as shown in Figure 1 were compared. Their geometrical parameters are listed in Table 1. Voltages were applied to the microheaters with an increment of 0.1 V at each measuring point until the liquid-metal heating channel broke. At each measuring point, the voltage was kept constant for 10 min to ensure that the microheater came to thermal equilibrium with surroundings.

Figure 2b,c plot the real-time temperature-voltage curves of the microheaters with and without the ventilating side-channels. In the case of 200 μm-wide liquid-metal heating channel, as seen in Figure 2b, the microheaters (1) with 10 μm-gap posts, (2) with 20 μm-gap posts, and (3) without the ventilating side-channels broke at 2.0, 1.7, and 1.5 V, respectively. 

The corresponding maximum temperatures that the microheaters can reach are 223.2, 169.5 and 124.7 °C, respectively. Similarly, in the case of 400 μm-wide liquid-metal heating channel as shown in Figure 2c, the microheaters broke at 1.5, 1.5 and 0.7 V, respectively, and the corresponding highest temperatures are 197.3, 177.1 and 68.5 °C, respectively.

In order to validate the effectiveness of the parallel ventilating side-channels, for each type of the microheater every experiment was performed for three times. The results are shown in Figure 3, which illustrates how the power and the resistance of the liquid-metal microheaters vary with temperature when increasing voltages were applied. As shown in Figure 3a,b, the microheaters with the ventilating side-channels can reach a higher temperature compared with those microheaters without the ventilating side-channels when supplied with the same power. This is because that the thermal conductivity coefficient of air (~0.026 W·m^−1^·K^−1^) is smaller than that of PDMS (~0.2 W·m^−^^1^·K^−^^1^) [22,23], and then the ventilating side-channels reduced heat dissipation and increased the temperature a little bit. Similarly, 200 μm-wide heating channels with the ventilating side-channels can reach a higher temperature with less power consumption than the same type of 400 μm-wide heating channels because the former has wider ventilating side-channels and smaller heat dissipation.

Figure 3c,d exhibit the relationship between the resistance and the temperature, which can be considered approximately segmented linear. For the 200 μm-wide liquid-metal heating channels (1) with 10 μm-gap posts, (2) with 20 μm-gap posts, and (3) without the ventilating side-channels, the slopes of their curves become steeper when temperature ranges from 105 °C to 120 °C, 105 °C to 125 °C and 75 °C to 90 °C respectively as seen in Figure 3c. Similarly, for the 400 μm-wide liquid-metal heating channels, the slopes of their curves become steeper when the temperature increases from 105 °C to 120 °C, 105 °C to 120 °C and 50 °C to 65 °C respectively as shown in Figure 3d. The significant increase of the resistance can be explained by the fact that voids occur within the EGaIn filled microchannels when the temperature reaches a critical value. The voids decrease the cross-section area of the liquid-metal heating channels, and thus increase the resistance as expressed in Equation (2) [18]:(2)$$R(T)=\frac{\rho (T)l(T)}{w(T)h(T)}=\frac{\rho_{0}(1+aT_{C})l_{0}(1+\beta T_{C})}{w_{0}(1+\beta T_{C})h_{0}(1+\beta T_{C})(1-r(T_{r}))}\;\frac{dR(T)}{dT}=\frac{1}{1+\beta T_{c}}\cdot \frac{\rho_{0}l_{0}}{w_{0}h_{0}(1-r(T_{r}))}\left(\alpha -\beta \frac{1+\alpha T_{c}}{1+\beta T_{c}}\right)$$
where *R(T)* is the resistance of the microheater, *T* is the current temperature of the microheater, *ρ*_0_ is the resistivity of the EGaIn at 20 °C, *T**_C_ = T* − 20, *l*_0_, *w*_0_, and *h*_0_ are the initial length, width, height of the liquid-metal heating channel at 20 °C respective,ly, *α* is the temperature coefficient of resistivity (TCR) of EGaIn, and *β* is the linear thermal expansion coefficient (TEC) of PDMS, *r*(*T_r_*) is reduction rate of the cross-section area of the liquid-metal heating channel due to voids, which is valid only when temperature exceeds the critical value *Tr* (*r(T_r_)* = 0 before voids occur). Herein, *α* and *β* are in the order of 10^−3^/°C and 10^−4^/°C, respectively, and assumed constant [24]. In the experiments, it was found that most voids occurred abruptly (approximately 30 ms) rather than forming gradually, so the relationship between the resistance and the temperature was approximately segmented linear. Table 2 listed some key indicators to compare the microheaters with different structures. The results clearly illustrate that the ventilating side-channels are able to increase the critical temperature at which voids occur as well as the highest temperature the microheaters can reach, but gaps between PDMS posts make no distinct difference in the performances of the microheaters before the microheaters were broken.

It is noted that, although there is such a relatively large increase in resistance(<5%, Figure 3c,d) near the critical temperature, the power still varied very smoothly in Figure 3a,b. The reason can be explained as follows. During the experiments, the voltage applied on the mircoelectrode was controlled to increase in steps all the time, as shown in Figure 2a,b. As the voltage increased from 0.0 V to about 2.0 V, the power applied on the heater would increase sharply with it. At such a strong increase of power input, the power change caused by the resistance change (<5%) will not be that obvious, so it seems that the power still increased smoothly when the voids occured, as shown in Figure 3a,b.

### 4.2. Analysis and Discussion

The formation and expansion of irregular voids inside the liquid-metal heating channel contributed to the heating channel breakage as mentioned in the Introduction. In order to reveal the reason why the voids occur and expand in the heated liquid metal, an experiment was performed as shown in Appendix A. As seen in the video, a voltage of 0.45 V was applied on the liquid metal channel. When the void occurred and began to grow at about 50 °C, the voltage was switched off and then an opposite voltage of −0.45V was applied on it immediately. As shown in Appendix A, the void continued to grow in the same direction rather than the opposite direction after the opposite voltage was applied and no obvious expansion of the PDMS microchannel was observed during the whole expansion. Because no obvious thermal expansion of the PDMS microchannel was observed and the expansion of EGaIn should be even smaller, the thermal expansion difference between the PDMS and EGaIn should not be the reason for the void growing in the video. Furthermore, if the electromigration was the main reason for the void growing, the growing direction should have been changed in the opposite direction when the voltage direction was changed. Thus, the reasons described in [18,19] should not be the main reason for this void growth. Considering the high thermal expansion rate of gas, the air trapped in the liquid metal might be the main reason for the void growing because the thermal expansion of the air could not be neglected when the Joule heat raised the temperature during the experiment. Thus, the reason of void generation and growing can be explained as follows: during the process of injecting the liquid metal, a small amount of air may remain inside the heating channel. Besides, air can also permeate through the PDMS and enter into the PDMS when pressed during the injection due to gas permeability of PDMS [25]. This amount of air may also release back into the heating channel when heated. When electric voltage is applied on the liquid metal, the air inside the channel will begin to expand because of increasing temperature, and finally causes the void to grow and liquid-metal heating to fail at the end. Figure 4a illustrates the whole process of the liquid-metal break-up in the heating channel, including voids occurring, growing and finally breaking the liquid metal. Figure 4b shows the final breaking of the liquid-metal heating channel induced by the voids. 

Figure 5a,b depict the formation process of voids during heating in the microchannel with ventilating side-channel. As shown in Figure 5b, when the temperature increased, voids occurred near the side-channel after 3 min of heating. Because the ventilating side channel can help to release the increasing air, the voids didn’t expand for the following 7 min. Therefore the ventilating side-channel enabled effectively to improve the working temperature range of the microheater with more application potential.

Based on the analysis above, the reason why the microheater with ventilating side-channels can reach higher temperature when heated can also be easily explained: (i) primary reason: the release of air trapped in the liquid-metal heating channel. Because the small gaps between PDMS posts give passages to release trapped air from the liquid-metal heating channel to the ventilating side-channels, the air can be instantly released instead of expanding when temperature increases; (ii) auxiliary reason: the formation of oxides of EGaIn. The gaps between the PDMS posts allow the air to contact with the liquid metal in the heating channel, therefore facilitating the formation of a protective layer of EGaIn oxides (mainly gallium oxide, Ga_2_O_3_) on the surface of the liquid metal [23]. This thin layer of EGaIn oxides can improve the mechanical stability of the liquid metal [6], and helps shape the liquid metal within the heating channel, which maintains the continuity of the EGaIn and therefore enables the heating channel with the ventilating side-channels harder to break.

### 4.3. Mechanical Tests

Pressing and bending tests were conducted on the microheaters with the ventilating side-channels as shown in Figure 6a because gaps between the PDMS posts might cause the liquid metal to overflow into the ventilating side-channels when the microheaters were subjected to large deformation. The resistance changes of the microheaters under increasing pressure are plotted in Figure 6b,c. To the 200 μm-wide heating channels with 10 and 20 μm gap posts, the maximum pressures that they can bear are 60.6 and 48.5 KPa, respectively. Similarly, the maximum pressures to the 400 μm-wide heating channels with 10 μm- and 20 μm- gap posts are 90.9 KPa and 54.6 KPa respectively. The maximal bending angle of all the miroheaters test is about 40° as shown in Figure 6a. The resistance values of the microheaters would significantly increase once applied pressure exceeded their maximum limits because EGaIn was pressured to overflow into the ventilating side-channels through the gaps.

Figure 6d,e plot the resistance change of the microheaters when bent. The four microheaters were subjected to 200 cyclic bendings, as shown in Appendix A. Equation (3) is used to measure the mechanical stability of the microheaters after the cyclic bendings:(3)$$\alpha =\frac{\left|r_{200}-r_{0}\right|}{r_{0}}\times 100\%$$
where *r*_200_ is the resistance of the heating channel after 200 cyclic bendings, *r*_0_ is the initial resistance of the heating channel. The values *α* of the 200 μm-wide liquid-metal heating channel with 10 μm- and 20 μm-gap posts are 0.82%, 2.09%, respectively. Similarly, those of the 400 μm-wide liquid-metal heating channel with the same posts are 0.41%, 15.71%, respectively. Except the microheater made of 400 μm-wide channel with 20 μm-gap posts, the other three microheaters exhibit good robustness for bending. These results show that the presence of the gaps between the PDMS posts possibly leads overflow of EGaIn into the ventilating side-channels when the microheaters are subjected to large deformation, and smaller gaps can prevent the problem.

During the mechanic tests, the liquid metal did not overflowed from main microchannel into the ventilating channels at beginning, as shown in Figure 7a–c. However, the electrode broke and the conductive liquid flowed into side-channel through gaps, which caused the cross-section of the electrode to shrink, resuling in the increase of the resistance. However, the microelectrode still shows good resistance properties to normal pressure and bending.

### 4.4. Heating Application

Due to the good flexibility and endurance of the novel microheaters at high temperatures, they were applied to heat the water in an aluminum-foil cup, as shown in Figure 8a. The microheater at the bottom is made of the two S-shaped heating channels parallel-connected in the circuit, whereas the other microheaters wrapping the sidewall of the aluminum-foil cup is made of three line-shaped parallel-connected heating channels. 50 mL water in the cup was heated from 27 °C to 55 °C in 30 min when electric voltages 3.2 V (1.38 A, *P* = 4.4 W) and 2 V (2.06 A, *P* = 4.1 W) was applied to the sidewall and the bottom microheater respectively as seen in Figure 6b. This water-heating test displays that the microheaters bent to flat and non-flat surfaces are capable of supporting currents up to 1.03 A (6.4 × 107 A·m^−2^, cross section: 80 μm × 200 μm) and 0.47 A (5.9 × 107 A·m^−2^, cross section: 80 μm × 100 μm) respectively without breaking down, hereby showing great potential use for soft and portable heating microdevices. The lamination of the soft heater is not good and the gap in between might greatly reduce the heating efficiency. That might explain why the demo heating took so long (30 min)

## 5. Conclusions

This study presented a novel liquid metal-based microheater with parallel ventilating side-channels. This soft microheater consists of a liquid metal-filled heating channel and two parallel ventilating side-channels. The liquid-metal heating channel is connected with the ventilating side-channels side by side by small gaps. Parametric experiments were conducted to compare the microheaters with and without the ventilating side-channels. The results showed that the microheaters with ventilating side-channels have higher working temperature in the same testing environment compared with those microheaters with no side-channels. To explain the improvement brought by the ventilating side-channels, we then designed an experiment to track the development of the voids within the liquid metal-heating channel during the heating. Based on that, a new hypothesis is proposed that thermal expansion of the trapped air leads to the formation and development of the voids, which finally break the liquid-metal heating channel. We attribute the improvement of the microheater with the ventilating side-channels to two reasons: (i) the gaps between PDMS posts provide trapped air passages to escape from the liquid-metal channel; (ii) air coming through the gaps facilitates the formation of EGaIn oxides on the liquid metal surface, and the oxides can improve the mechanical stability of the liquid metal within the heating channel. Afterward, the microheaters with the ventilating side-channels are subjected to pressing and bending to quantify their mechanical stability. The results show gaps between the PDMS posts possibly lead to easier overflow of EGaIn into the ventilating side-channels and smaller gaps can improve mechanical stability. Finally, the new microheaters are applied to heat water from 27 °C to 55 °C in 30 min by attaching them to the bottom and the curved sidewall surface of a cup. We believe the novel microheater can be applied to a large variety of wearable and medical instruments such as heating soles, implantable heating probes, PCR, etc., and the structure of one liquid-metal heating channel sandwiched between two parallel ventilating side-channels offers us an effective way to fabricate liquid-metal wires with fine electric stability.

## 6. Experimental Section

### 6.1. Synthesis of liquid PDMS

The liquid PDMS was prepared by mixing pre-polymer and curing agent at a weight ratio of 10:1 using a glass rob for 10 min. After that, this mixture was vacuum evacuated for 1 h.

### 6.2. Preparation of Eutectic Alloy Ga_75.5_In_24.5_


Gallium and indium were first melted at 150 °C at a weight ratio of 75.5:24.5 in a dry box for 2 h. We then stirred them using a glass rod for 10 min until all the metal was blended. Afterward, the metal was placed back into the dry box for 30 min. The liquid metal Ga_75.5_In_24.5_ was finally fabricated after the alloy was cooled down to room temperature.

## Figures and Tables

**Figure 1 micromachines-11-00133-f001:**
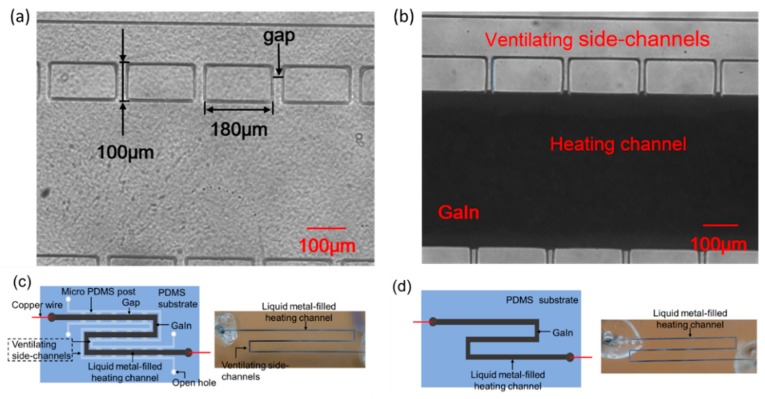
Microstructure of the soft microheater (**a**) before and (**b**) after injectinge GaIn. S-shaped liquid metal-based microheater (**c**) with and (**d**) without the parallel ventilating side-channels.

**Figure 2 micromachines-11-00133-f002:**
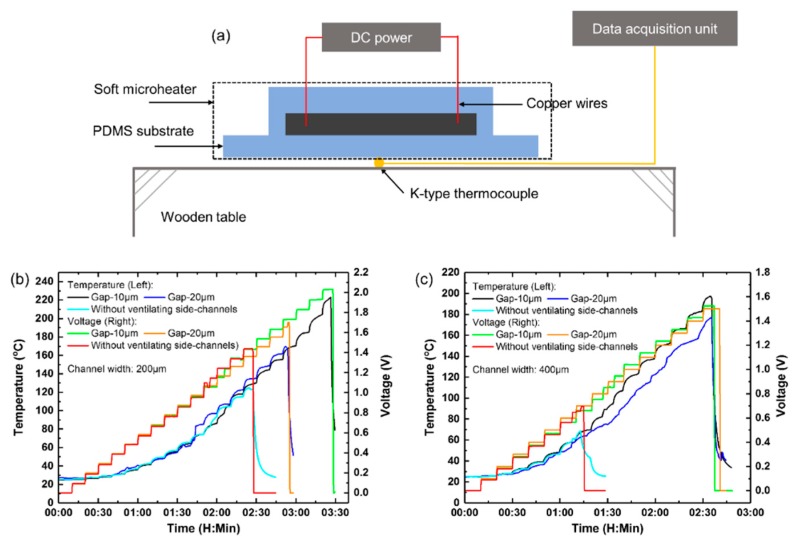
(**a**) Schematic experimental setup for Joule-heat generation and data acquisition. Real-time temperature-voltage characteristic curves of the microheaters with and without the parallel ventilating side-channels: (**b**) 200 μm-, (**c**) 400 μm-wide liquid-metal microchannels.

**Figure 3 micromachines-11-00133-f003:**
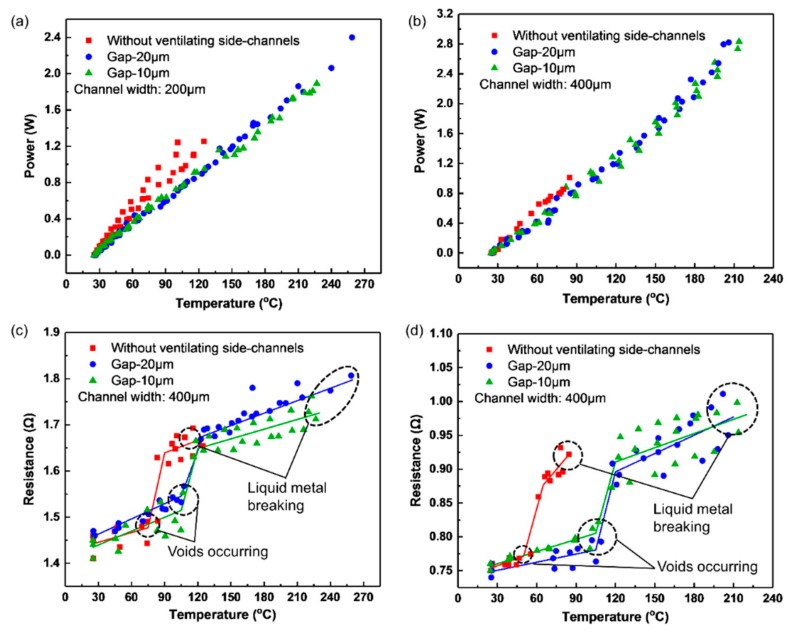
Power of the microheaters with (**a**) 200 μm- and (**b**) 400 μm-wide liquid-metal heating channel as functions of temperature; Resistance of the microheaters with (**c**) 200 μm- and (**d**) 400 μm-wide liquid- metal heating channel as funtions of temperature. Three repeated measurements are shown.

**Figure 4 micromachines-11-00133-f004:**
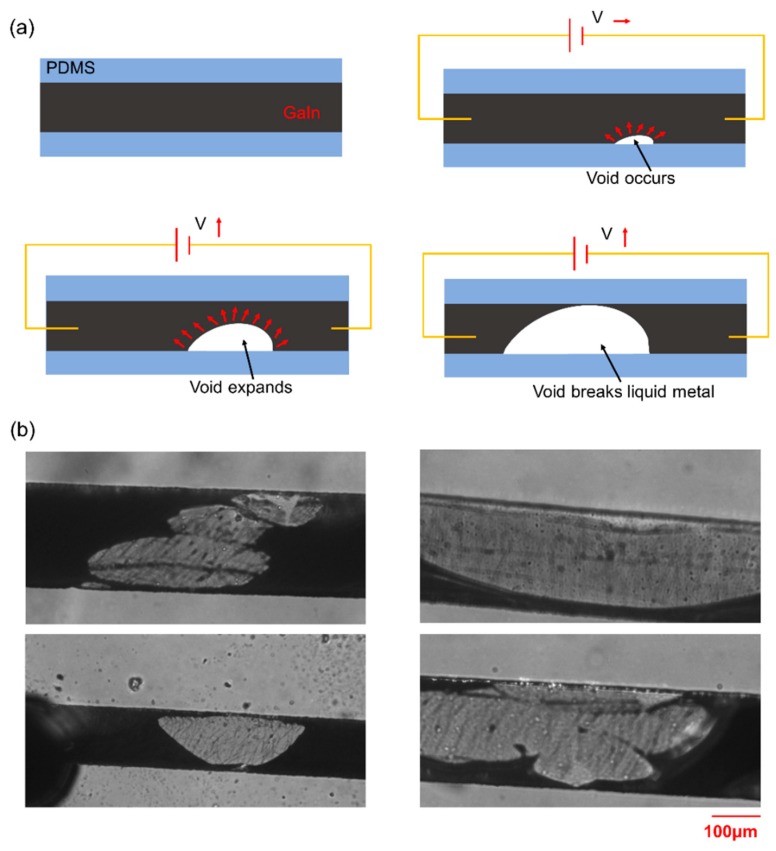
(**a**) Whole process of the liquid-metal break-up within the heating channel induced by voids. (**b**) Microscopic view of the final liquid-metal break-up within the heating channel.

**Figure 5 micromachines-11-00133-f005:**
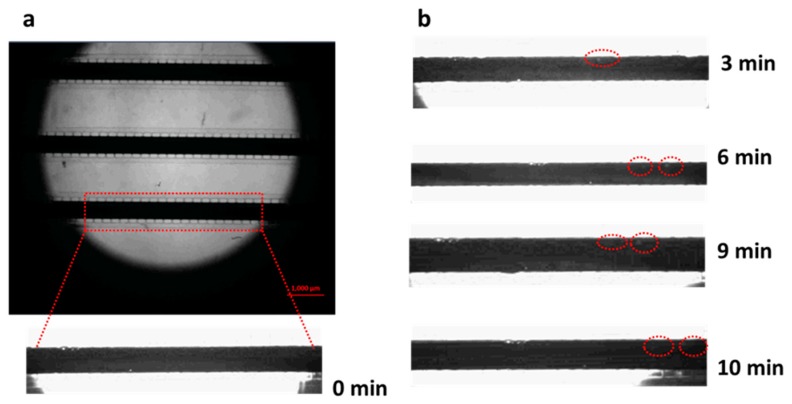
The growth of voids in microchannel with ventilating gaps. (**a**) depicts the morphology of the microelectrode without heating. The inset shows a segment of electrode at the bottom. (**b**) records the morphological changes after working 3, 6, 9 and 10 min of heating. With the increase of temperature, the voids (red dot ellipse) started to birth but not expend.

**Figure 6 micromachines-11-00133-f006:**
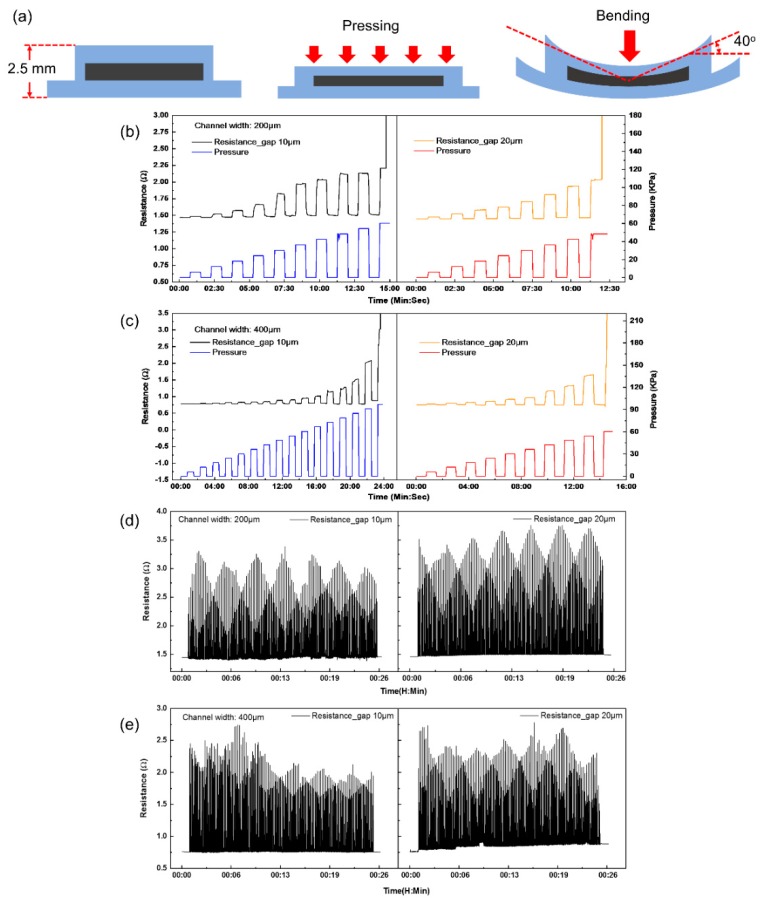
(**a**) Microheaters are subjected to pressing and bending. Resistance change of the microhearers with (**b**) 200 μm-, (**c**) 400 μm-wide liquid-metal heating channel under increasing pressure. Resistance change of the microhearers with (**d**) 200 μm-, (**e**) 400 μm-wide liquid-metal heating channel under cyclic bending.

**Figure 7 micromachines-11-00133-f007:**
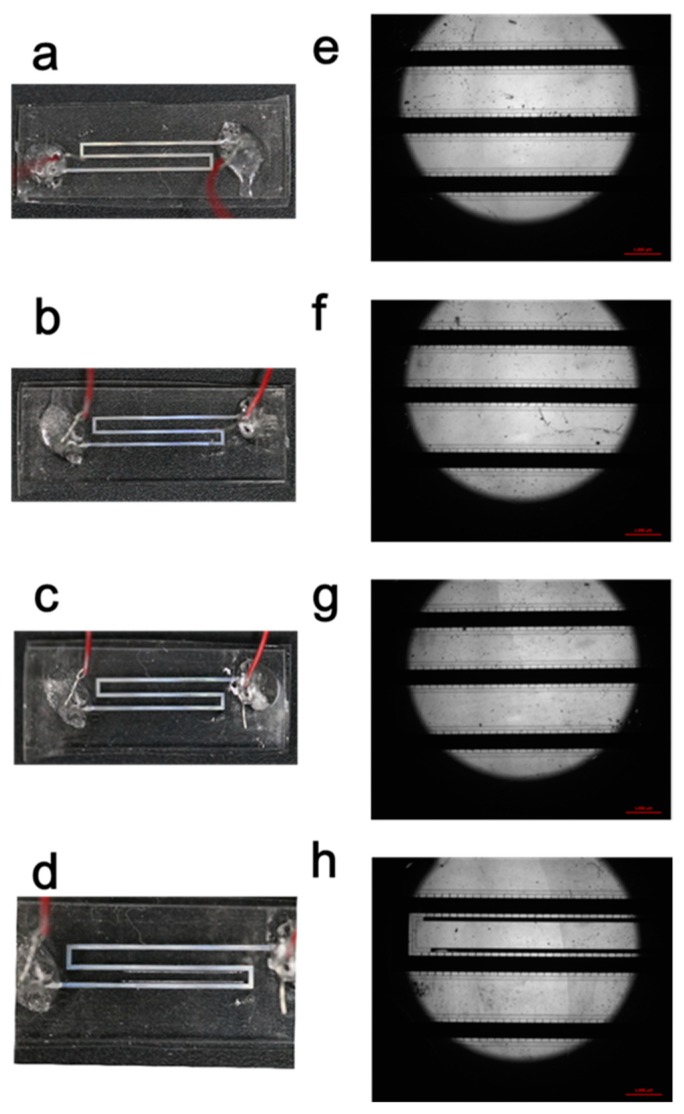
The morphology of 400 μm-wide microeletrode with 20 μm-gap posts after considerable bending cycles. (**a**–**d**) displays the state of the microelectrode, as carried out 0, 50, 100, 200 cycles; (**e**–**h**) representing the microscopic image respectively correspond to (**a**–**d**).

**Figure 8 micromachines-11-00133-f008:**
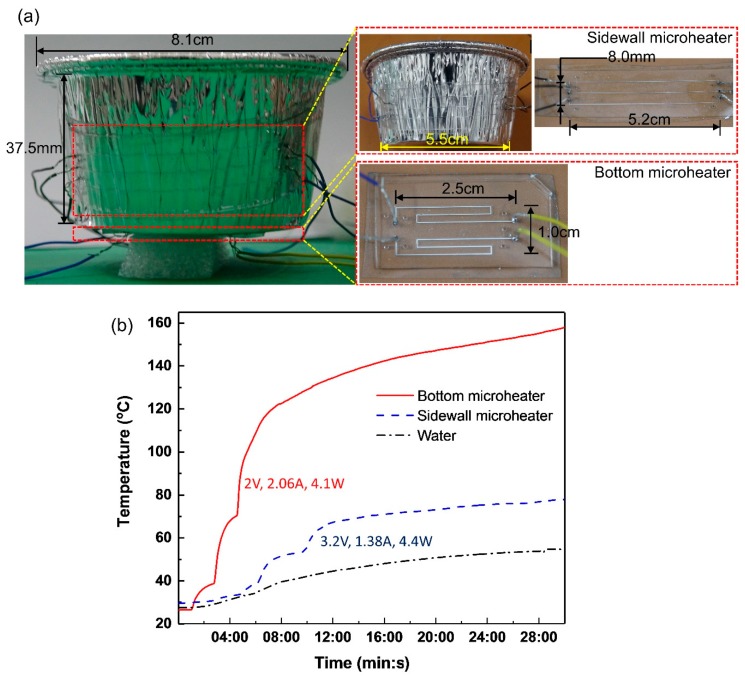
(**a**) Two microheaters are attached to the bottom and the sidewall of an aluminum-foil cup respectively. (**b**) Real-time temperature change of the bottom microheater, the sidewall microheater and the water in the cup.

**Table 1 micromachines-11-00133-t001:** Dimensions of microheaters with and without parallel ventilating side-channels.

Parameters	With Ventilating Side-Channels				Without Ventilating Side-Channels	
Width of liquid metal channel (μm)	200	200	400	400	200	400
Height of liquid metal channel (μm)	80	80	80	80	80	80
Length of liquid metal channel (cm)	6	6	6	6	6	6
Gap between PDMS posts (μm)	10	20	10	20	N/A	N/A
Width of ventilating side-channels	200	200	100	100	N/A	N/A
Initial resistance (Ω)	1.45	1.46	0.76	0.76	1.41	0.75

**Table 2 micromachines-11-00133-t002:** Comparison between microheaters with and without parallel ventilating side-channels.

Width of Heating Channels (μm)	Cap between Post (μm)	Critical Temperature (°C)	Highest Temperature (°C)	Corresponding Power (w)
200	10 20 No post	−105 −105 −75	223.40 212.80 113.70	1.89 2.40 1.25
400	10	−105	207.9	2.83
	20 No post	−105 −50	195.00 77.18	2.82 1.01

Critical temperature: temperature point where voids start to birth; Highest temperature: the highest temperature the microheaters can reach.

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
