# Peer review of "A Performance-Enhanced Liquid Metal-Based Microheater with Parallel Ventilating Side-Channels"

_micromachines, 2020, doi:10.3390/mi11020133_

Round 1

Reviewer 1 Report

In this paper, the authors present a durable and easy-to-fabricate flexible micro-heater based on liquid-metal infused microchannels. They show that the previous maximum temperature of 100°C can be doubled by introducing side channels which can “absorb” the generated gas and hence reduce the breakdown of the liquid metal plug. The authors also elucidate the fundamentals underlying void formation and show that existing theories cannot accurately predict void formation. Instead, they propose that entrapped air expands and/or permeates through the PDMS and ultimately leads to critical void formation.

Overall, this is a mostly well written paper which has the potential to advance flexible heaters and MEMS. Few minor questions/comments:

All gallium alloys oxidize readily in the presence of miniscule amounts of O2. This oxide layer usually makes the liquid metal “sticky” (instead of the high mobility of pristine Ga alloys). Can the authors please comments on the influence of such oxide layer during filling? Check line 173 – something doesn’t look right there… In line 179/180 the authors claim that the gap size between the posts does not significantly influence in the thermal behavior of the heaters, yet Fig. 2b looks like the 10 um gaps can reach significantly (!) higher temperatures than the 20 um gaps! Please clarify! Table 2: how is the critical temperature defined & derived from the temperature data? Along the same lines, I am a little surprised that there is such a strong increase in resistance, but not in the power for a given temperature increase in Fig. 3? Please clarify.

Author Response

Response to reviewers’ comments:

In this paper, the authors present a durable and easy-to-fabricate flexible micro-heater based on liquid-metal infused microchannels. They show that the previous maximum temperature of 100°C can be doubled by introducing side channels which can “absorb” the generated gas and hence reduce the breakdown of the liquid metal plug. The authors also elucidate the fundamentals underlying void formation and show that existing theories cannot accurately predict void formation. Instead, they propose that entrapped air expands and/or permeates through the PDMS and ultimately leads to critical void formation.

Overall, this is a mostly well written paper which has the potential to advance flexible heaters and MEMS. Few minor questions/comments:

Q1: All gallium alloys oxidize readily in the presence of miniscule amounts of O2. This oxide layer usually makes the liquid metal “sticky” (instead of the high mobility of pristine Ga alloys). Can the authors please comments on the influence of such oxide layer during filling?

Reply: Thanks for the professional comments. Before the filling process, we treated the alloy with NaOH solution to remove the oxide. During the filling process, the liquid metal is always in a moving state and the filling process was not long, so the oxidation layer cannot be fully generated during that process. Therefore, during our experiment oxidization has little effect on the filling process.

Q2: Check line 173 – something doesn’t look right there… In line 179/180 the authors claim that the gap size between the posts does not significantly influence in the thermal behavior of the heaters, yet Fig. 2b looks like the 10 um gaps can reach significantly (!) higher temperatures than the 20 um gaps! Please clarify! 

Reply : Thanks for the professional comments. We meant that “gap size makes no distinct difference in the performances of the microheaters before the microheaters were broken.” We added “before the microheaters were broken” at the end of the description to clarify the meaning (See line 180). Thanks for the helpful suggestion. It makes the manuscript much more professional.

Q3: Table 2: how is the critical temperature defined & derived from the temperature data? 

Reply:Thanks for the professional comments. The critical temperature means the temperature at which voids just start to form.The critical temperature is defined in paper to depict the theraml behaviors of microheaters and acts as a significant property index of microheaters. The related information about cirtical temperture has been described in the revised manuscript (see line 196 )

Q4: Along the same lines, I am a little surprised that there is such a strong increase in resistance, but not in the power for a given temperature increase in Fig. 3? Please clarify.

Re: Thanks for the professional comments. Just as the reviewer said, although there is such a relatively large increase in resistance(<5%, Figure 3c and 3d) near the criticle temperature, the power still varied very smoothly in Figure 3a and 3b. The reason can be explained as follows. During the experiments, the voltage applied on the mircoelectrode was controlled to increase in steps all the time, as shown in Figure 2 (a) and (b). Voltage increasing from 0.0V to about 2.0V, the power applied on the heater would increase sharply with it. At such a strong increase of power input, the power change caused by the resistance change (<5%) will not be that obvious. So it seems that the power still increased smoothly when the voids occured as shown in Figure 3a and 3b. (See in line 183-190).

Reviewer 2 Report

The submitted manuscript presents a study to understand and solve an issue found in soft microheaters made of liquid metals. Overall, the paper is very well written. The authors present their idea clearly and design experiments to verify and support their ideas with thoughtful considerations. I recommend it for publication after minor revision. My comments are listed below:

The authors propose to use side channels with venting gaps to release the air/void bubble generated during heating. The results of this idea need to be strengthened as there is no experimental image or videos to support this design in the current manuscript. The authors described how other existing explanations are not valid. It is necessary for the authors to show images (i.e., a series of snapshots) to demonstrate the function of the air releasing from their side channel design. Putting the video of this process in the supplementary will help the reader to easily understand their clever design. The mechanical reliability tests of the proposed device need some more detail or improvements. For example, the authors can provide some pictures depicting the failure of the device such as the overflow of liquid metals to the side channel under high pressure and how does the 400-micron channel look like after 200 cycles with 15% resistance change. The bending test should provide some more quantifiable parameters for such characterization. For example, what is the minimum curvature of the liquid metal channel during the bending tests? Figure 2 (b) and (c) might better be presented in “current” rather than “voltage” because the authors are providing voltage to the heater and even if the liquid metal is broken, the voltage (i.e., open-circuit voltage) should be maintained. The authors need to provide the thickness of the PDMS device. Currently, only the liquid metal channel info has been provided but the heating is actually provided through the PDMS and the thickness of the PDMS matters whether the measurement is performed in the steady-state. The heater demo should not be done with an aluminum-foil cup which is very rough and uneven. The lamination of the soft heater is not good and the gap in between might greatly reduce the heating efficiency. That might explain why the demo heating took so long (30 minutes). Some more minor issues/typos: Line 8-9, email address of all authors should not appear in affiliation line #1 Figure 1 (a) and (b) should just provide one set of channels. The current images shown with the dark shadow provide insufficient resolution and do not look professional. Line 175, please provide a time scale of the formation of the voids, in seconds, milliseconds, …? Table 2 needs better formatting to clearly show the 200-micron vs. 400-micron devices. Line 247, equation 3 should be “times 100%”?

Author Response

Response to reviewers’ comments:

The submitted manuscript presents a study to understand and solve an issue found in soft microheaters made of liquid metals. Overall, the paper is very well written. The authors present their idea clearly and design experiments to verify and support their ideas with thoughtful considerations. I recommend it for publication after minor revision. My comments are listed below:

1 The authors propose to use side channels with venting gaps to release the air/void bubble generated during heating. The results of this idea need to be strengthened as there is no experimental image or videos to support this design in the current manuscript. . The authors described how other existing explanations are not valid. It is necessary for the authors to show images (i.e., a series of snapshots) to demonstrate the function of the air releasing from their side channel design.

Reply : Thanks for the professional comments. We highly agree with the reviewer’s comments. To capture the voids generation and expansion for the proposed microheater, we performed several new experiments and fortunately captured one as shown in Figure 5 in the revised manuscript. The Figure 5a and 5b depict the forming process of voids during heating in the microchannel with ventilating side-channel. As shown in Figure 5b, when the temperature increased, voids occurred near the side-channel after 3 minutes of heating. Because the ventilating side channel can help to release the growing air, the voids didn’t expand for the following 7 minutes. (See line 226-232)

2 The mechanical reliability tests of the proposed device need some more detail or improvements. For example, the authors can provide some pictures depicting the failure of the device such as the overflow of liquid metals to the side channel under high pressure

Reply : Thanks for the professional comments. We highly agree with the reviewer’s comments. We have added some images to depict the mechanical reliability test in revised manuscript (see the Figure 7 in the page of 11).

3 how does the 400-micron channel look like after 200 cycles with 15% resistance change.

Reply : Thanks for the professional comments. Please see the Figure 7d and 7h in the revised manuscript, it shows the morphology of 400-micron channel after 200 cycles of bending.

4 The bending test should provide some more quantifiable parameters for such characterization. For example, what is the minimum curvature of the liquid metal channel during the bending tests?

Reply : Thanks for the professional comments. The compact bending tests are carried out with help of the stepper motor. Limited by the experimental devices, the tests did not choice more rational methods, such as bending test measured by angle. But the experiments we had carried out also scientifically depicted the good mechanic properties and flexibility of the microheater. Noticeably, the maximum bending angle up to about 40° in tests. The related description has been added in line 266 in the revised manuscript.

5 Figure 2 (b) and (c) might better be presented in “current” rather than “voltage” because the authors are providing voltage to the heater and even if the liquid metal is broken, the voltage (i.e., open-circuit voltage) should be maintained.

Reply : Thanks for the professional comments. We used the “voltage” to present the Figure 2, because that is how we made the experiments. We kept the voltage constant for 10 mins and then increased it for 0.1V every time. Because the resistant will always change with the temperature, if we use “current” to present Figure 2, the current will keep changing during the whole experiment and the reader will not exactly know how we control the heating directly by Figure 2.

6 The authors need to provide the thickness of the PDMS device. Currently, only the liquid metal channel info has been provided but the heating is actually provided through the PDMS and the thickness of the PDMS matters whether the measurement is performed in the steady-state.

Reply: Thanks for the professional comments. We highly agree with it. The thickness of the soft microheater (2100 μm) has been mentioned in the revised manuscript (see the line 113,114 ) .

7 The heater demo should not be done with an aluminum-foil cup which is very rough and uneven. The lamination of the soft heater is not good and the gap in between might greatly reduce the heating efficiency. That might explain why the demo heating took so long (30 minutes)

Reply: Thanks for the professional comments. We highly agree with it. To find out the real application effect, we randomly chose such a cup with rough and uneven surface from the supermarket and used our heater to heat it. Although the heating time is so long, it shows us a potential real application. We believe by applying more heaters the heating time may be greatly shortened, but that has go beyond the scope of this work.

8 Some more minor issues/typos: Line 8-9, email address of all authors should not appear in affiliation line #1 Figure 1 (a) and (b) should just provide one set of channels. The current images shown with the dark shadow provide insufficient resolution and do not look professional. Line 175, please provide a time scale of the formation of the voids, in seconds, milliseconds, …? Table 2 needs better formatting to clearly show the 200-micron vs. 400-micron devices. Line 247, equation 3 should be “times 100%”? 

Reply: Thanks for the professional comments. According to the reviewer’s suggestion we have modified our manuscript. The time scale (30ms) of the formation of the voids has been added in the revised manuscript (see line